# Thermal Tolerance of Fruit-Feeding Butterflies (Lepidoptera: Nymphalidae) in Contrasting Mountaintop Environments

**DOI:** 10.3390/insects11050278

**Published:** 2020-05-01

**Authors:** Vanessa Diniz e Silva, Marina Vale Beirão, Danon Clemes Cardoso

**Affiliations:** 1Programa de Pós-graduação em Ecologia de Biomas Tropicais, Universidade Federal de Ouro Preto, Minas Gerais 35400-000, Brazil; vanessa.diniz@aluno.ufop.edu.br (V.D.eS.); marinabeirao@ufop.edu.br (M.V.B.); 2Departamento de Biodiversidade, Evolução e Meio Ambiente, Universidade Federal de Ouro Preto, Minas Gerais 35400-000, Brazil

**Keywords:** CTmax, campo rupestre, forest islands, climate change, Espinhaço range, Brazil

## Abstract

Ectothermic organisms, such as insects, are highly temperature dependent and are good models for studies that predict organisms’ responses to global climate change. Predicting how climate change may affect species distributions is a complicated task. However, it is possible to estimate species’ physiological constraints through maximum critical temperature, which may indicate if the species can tolerate new climates. Butterflies are useful organisms for studies of thermal tolerance. We tested if species have different thermal tolerances and if different habitats influence the thermal tolerance of the butterflies present in Brazil’s campo rupestre (open areas) and forest islands (shaded areas). A total of 394 fruit-feeding butterflies, comprising 45 species, were tested. The results separated the species into two statistically different groups: the resistant species with maximum critical temperature of 53.8 ± 7.4 °C, and the non-resistant species with maximum critical temperature of 48.2 ± 7.4 °C. The species of butterflies displayed differences in maximum critical temperature between the campo rupestre and forest islands that can be related to the two distinct habitats, but this did not correlate phylogenetically. Species from the forest islands were also divided into two groups, “resistant” and “non-resistant”, probably due to the heterogeneity of the habitat; the forest islands have a canopy, and in the understory, there are shaded and sunny areas. Species from forest islands, especially species that displayed lower thermal tolerance, may be more susceptible to global warming.

## 1. Introduction

Ectotherms, such as insects, have physiological characteristics that make them temperature dependent [1]. This limitation makes them good models for studies that seek to predict organismal responses to possible global climate change and habitat choice. Insects are adapted and acclimated to specific temperature ranges, and changes in temperature affect the performance, phenology, and distribution of the species [2,3]. Further, examining physiological and ecological traits in a phylogenetic context provide valuable information about their modes of evolution. Temperature seems to be a key factor structuring the biological community, mainly in habitats with extreme variation and fluctuations during the day, or between seasons [4].

Predicting how climate change can affect species distributions constitutes a significant challenge for contemporary ecology [5]. Additionally, estimating the temperature limits that species can tolerate in several microhabitats is a difficult task [6]. However, it is possible to estimate the thermal tolerance of a species by measuring their maximum critical temperature (CTmax) [7,8]. Information about the CTmax of organisms can be especially useful for understanding the thermal response of thermally restricted species, such as butterflies, facing climate change.

Butterflies are very sensitive to abiotic conditions such as temperature, humidity, and presence of shades or light [9], as well as the biotic, because they depend on the presence of their host plants. Consequently, closely related to their habitat, and since they respond rapidly to changes in the environment, they are a useful bioindicator organism [9,10,11]. Thus, different habitats are expected to contain different butterfly species.

In the mountaintops of southeastern Brazil, there are many habitat types, from an open environment like the campo rupestre [12] to the forest islands (regionally known as capão de Mata) [13]. The campo rupestre is composed of herbs and small shrubs associated with poor soils and the presence of exposed rocks [14,15]. The vegetation of the forest islands is formed of trees from the Atlantic Forest and the Cerrado; thus, the area is always humid and is composed of plants with a long life cycle [13]. The main distinction between these two habitats is the soil. In campo rupestre, bare rocks are exposed, so the ground experiences rapid drainage and leaching and, in some parts, organic matter accumulates and forms a substrate for shrubs [12]. Conversely, in the forest islands, the soil is rich and can sustain trees with a canopy of 12 to 16 m high [13]. The consequence of this difference in the soil has led to a differentiation of vegetation, leading to a differentiation of habitat, one a grassland and the other forest. This makes the campo rupestre and the forest islands two completely different habitats, one more exposed to the sun, winds, and dryness, and the other with less variation in temperature and humidity. This mountain range in Brazil is remarkable for having two separate hotspots: the Cerrado and Atlantic Forest [16]. Such areas are useful settings in which to test community thermal conditions because both occur at the same altitude, in adjacent areas, and are on the same slope of the mountain.

The main objective of the present study was to test, using CTmax, if fruit-feeding butterfly species show different thermal tolerances across habitats. We expect that the butterfly species of the campo rupestre exhibit higher thermal tolerance values than the butterfly species of the forest islands, because the temperature is higher and the sun incidence more prominent in campo rupestre. So, we tested if the thermal tolerance of fruit-feeding butterflies is intrinsic to the species and if the temperature of the habitat can limit the dispersion and the establishment of some species. Moreover, we explored two other hypotheses: (i) the thermal tolerance of the fruit-feeding butterfly species of the same habitat are similar; and (ii) there is a relationship between the CTmax and habitat occurrence of fruit-feeding butterflies resulting from shared ancestry.

## 2. Materials and Methods

### 2.1. Study Area

This study was carried out in Serra do Ouro Branco State Park, located in Ouro Branco municipality, in the southern portion of the Espinhaço mountain range, southeastern Brazil (Figure 1). The Espinhaço mountain range is the second biggest mountain range in South America [15,16]. The climatic regime is characterized as mesothermic, (Cwb) according to Köppen’s classification, with dry winters and rainy summers [17]. Mean annual temperature is 20.7 °C. Mean annual rainfall is ca. 1,188,200 mm. July to September are the driest months, and November to February, the wetter months.

Two habitats are more common at the top of the mountain in Ouro Branco State Park (ca. 1400 m a.s.l.): the campo rupestre and the forest islands [18]. The campo rupestre vegetation is characterized by herbaceous species, with high levels of endemism, and consequently it has a unique species composition [12]. The forest islands, on the contrary, have a similar floristic composition to the semi-deciduous forests associated with the Atlantic Forest domain.

### 2.2. Sampling Design

We sampled fruit-feeding butterflies from three sites in the campo rupestre (open) and three sites in the forest islands (shaded). The sites were at least 500 m from each other. At each site, five butterfly traps were installed at a distance of 50 m from each other in a line, and with the base about 1 m above the ground, in total there were 30 traps. In the forest islands, the traps were hung more in the middle of the fragment, to avoid the edge, and the trap line followed the shape of the fragment. The butterflies were collected with attractive bait traps using fermented banana. The traps were inspected every day during the sampling period. All captured butterflies were marked with an identification number and kept in a glass container (52 × 27 × 39 cm) without the top, and closed with a thin fabric and with some cotton emerged with water in the bottom to reduce stress during transportation to the laboratory to carry out the thermal tolerance tests. The number of the butterflies was related to the date of capture, the number of the trap, the fragment, and the habitat. Fieldwork was conducted monthly over a year (December 2017 to November 2018) in five days of sampling.

### 2.3. Thermal Tolerance Test

To test thermal tolerance, the butterflies, alive and with normal behavior (not too quiet), were acclimatized for 30 min at 21 °C. Then, each individual was placed in a 2 L glass beaker that was submerged in a water bath (Quimis; Model: G215M1; Precision: 0.05 °C) [19] with initial temperature 21 °C. The temperature inside the glass was tracked using a thermometer. When the butterflies lost motor control (usually not staying up, falling aside), the temperature of the air inside the glass was recorded and this was considered the CTmax [3,8] of the individual tested. After the test, the butterflies were identified using field guides [20,21] and they were deposited in the collection of the Universidade Federal de Ouro Preto.

### 2.4. Statistical Analysis

To test if the thermal tolerance of butterfly species and the habitat differed, we built a linear mixed model (LMM) with the CTmax of each tested specimen (response variable) and the species, the habitat, and the interaction between them (explanatory variable). Because we tested some species more than others, we inserted the species as the random variable. For this model, we only used species with more than three individuals tested for CTmax. We built a null model to contrast the first model. If the models differed, we assumed that the thermal tolerance was different in at least one of the response variables. For the model simplification, we first extracted the interaction and then contrasted with our first model. If significant, the first model was considered the final model, and if not significant, we assumed the minimal model without the interaction. For the mixed model, we used the package lme4 [22].

Finally, to evaluate if thermal tolerance in each habitat was similar among the species, we built two generalized linear models (GLMs), one for each habitat (the campo rupestre and the forest islands). We separated all species that were tested at least three times from each habitat. We tested the CTmax (response variable) and the species that occurred in the habitat (explanatory variable). If the models were significant, we did a contrast analysis to determine in how many groups the species were separated and which species belonged to each group. The nature of the data meant that all models presented normal distribution. All analyses were performed in R v 1.1.447 [23].

### 2.5. Phylogenetic Correlation

In order to test the correlated evolution of CTmax and habitat occurrence in Nymphalidae, we estimated a phylogenetic hypothesis of the phylogenetic relationships of butterflies that significantly differed in relation to CTmax and habitat occurrence. We did a preliminary search in GenBank to look for a molecular marker that covers most of the sampled species within each group. From this, we were able to conclude that the *Cox* I (COI) gene would allow us the most inclusive study; it is also the most frequently sequenced mitochondrial marker studied in insects. Indeed, COI is the widely adopted mitochondrial marker used as a barcode sequence. Then, using a matrix of 1500 base pairs with the Kimura 2-parameter model, we constructed a neighbor joining tree covering 21 species comprising the CTmax groups, which were separated significantly between habitats. Statistical support was assessed with bootstrap resampling over 1000 replications. The GenBank accession numbers for the species sequences reported in this paper are presented at the tips of the phylogenetic tree.

We evaluated the phylogenetic signal using a comparative phylogenetic approach, with Pagel’s λ [24], Blomberg’s *K* [25], Abouheif’s *C*_mean_ index [26], and Moran’s I [27] compared using the phytools, picante, and adephylo packages in R. The first two tests assumed a Brownian motion model of evolution whereas the last two were autocorrelation indices, not based on an explicitly evolutionary model [28]. Blomberg’s *K* is a variance ratio where the variables tested are independent from the phylogeny when *K* < 1, and dependent on the phylogeny when *K* ≥ 1. Pagel’s λ is a scaling parameter ranging from 0 (complete phylogenetic independence) to 1 (phylogenetic dependence). However, Moran’s I equals 0 indicates that species in the phylogeny resemble each other as expected by the Brownian motion model and, when lower than 0, species resemble one another less. However, when I is higher than 0, the model suggests that closely-related species in phylogeny resemble each other in relation to the studied trait. Abouheif’s *C*_mean_ is similar to Moran’s I, except that it does not display a 0 *C*_mean_.

In order to determine the relationship between physiological and ecological traits, we employed a phylogenetic generalized least squares (PGLS) analysis with ANOVA, taking CTmax as the continuous trait, and habitat occurrence (forest islands and the campo rupestre) as the categorical trait. All analyses were performed in R with the packages ape [29] and phytools [30].

## 3. Results

We tested the thermal tolerance of 394 butterflies, comprising 45 species in the Nymphalidae family. Of the total sampled species, 28 species were collected from the campo rupestre and 34 from the forest islands. The most abundant species in the campo rupestre was *Yphthimoides patrícia* (34, 41.4%), and in the forest islands, the most abundant species was *Godartiana muscosa* (94, 30.1%) (Table 1). The CTmax of the species and the habitats differed from the null model (*p* = 0.005; Χ^2^ = 67.98). The interaction between the species and the habitat was not significant, indicating that the pattern of the temperature CTmax of the species remains the same between habitats. The random effect explained only 10.39 (± 3.223) of the variance, and the residual (habitat) explained 47.22 (± 6.872).

The species separated into two groups according to their thermal tolerance (n = 26 species with more than three individuals, *F* = 3.0, *p* < 0.001). The species *Carminda paeon, Eryphanis reevesii, Eunica maja, Hermeuptychia* sp1, *Memphis otrere, Yphthimoides pacta, Y. patricia*, and *Yphthimoides renata* (53.8 ± 7.4 °C) displayed higher temperature tolerance than the species *Archaeoprepona amphimachus*, *Blepolenis batea*, *Caligo arisbe*, *Dasyophthalma rusina*, *Epiphile orea*, *Forsterinaria necys*, *Forsterinaria quantius*, *G. muscosa*, *Moneuptychia itapeva*, *Moneuptychia giffordi*, *Morpho helenor*, *Opsiphanes invirae*, *Pharneuptychia pharnabazos*, *Taygetis drogoni*, *Taygetis laches*, *Yphthimoides affinis*, *Yphthimoides borasta,* and *Yphthimoides straminea* (48.2 ± 7.4 °C, Figure 2). The butterflies from the campo rupestre (52.7 ± 8.4 °C) displayed a higher thermal tolerance than the butterflies from the forest islands (49.3 ± 7.1 °C), when considering abundance (*F* = 13.3, *p* < 0.001; Figure 3).

When we separated species by habitat, there were more tested species in the forest islands. Only five species occurring in the campo rupestre were tested more than three times (*E. orea, G. muscosa, M. itapeva, Y. pacta,* and *Y. patricia*) and their thermal tolerance was similar, with an average temperature of 54.6 ± 8.7 °C (*F* = 1.2, *p =* 0.312). In the forest islands, there was a different pattern; first, there were twenty species tested more than three times in this habitat (Table 1), and second, the species divided into two groups (*F* = 3.3, *p* < 0.001). The species *C. paeon, E. reevesii, E. maja, Hermeuptychia* sp1, *M. otrere, P. pharnabazos, T. drogoni, Y. renata*, and *Y. straminea* represent the group with the higher mean CTmax (52.8 ± 7.0 °C), whereas the species *C. arisbe*, *D. rusina*, *F. necys*, *F. quantius*, *G. muscosa*, *M. giffordi*, *M. helenor*, *O.s invirae*, *T. laches*, *Y. affinis,* and *Y. borasta* represent the group with the lower mean CTmax (47.4 ± 6.5 °C; Figure 4).

The phylogenetic hypothesis based on the COI barcode is shown in Figure 5; the topology is in agreement with recent butterfly phylogenies. All indices indicated a lower or non-existent phylogenetic signal for CTmax and habitat occurrence. Only Blomberg’s *K* showed significant phylogenetic dependence for CTmax (*K* = 1.02, *p =* 0.023) and habitat (*K* = 1.02, *p =* 0.023), while Pagel’s λ indicated a correlation for CTmax (λ = 0.55, *p* = 0.158) and habitat (λ = 0.99, *p =* 0.23), but this was not significant. Likewise, Abouheif’s *C*_mean_ for CTmax (*C*_mean_ = 0.23, *p =* 0.05) and habitat (*C*_mean_ = 0.24, *p* = 0.065) and Moran’s I for CTmax (I = 0.19, *p =* 0.06) and habitat (I = 0.20, *p =* 0.071) also indicated no autocorrelation of CTmax and habitat with the phylogenetic relations of the species studied. Phylogenetic ANOVAs demonstrated that CTmax is not a good predictor of habitat among Nymphalidae butterflies in mountaintops (F_(2,21)_ = 0.5020, *p =* 0.487).

## 4. Discussion

In this study, we showed that species of fruit-feeding butterflies could be divided into two groups according to thermal tolerance. Further, the butterfly community from the campo rupestre displayed higher thermal tolerance than butterflies from the forest islands when considering the species abundances. Yet, taking into account the average maximum temperature of each species, the thermal tolerance between the habitats was similar. We suggest that the thermal susceptibility of the butterflies is intrinsic to the species, and habitat type can operate as a filter on selection of the community. Our phylogenetic comparative methods approach also indicates that CTmax is more related to each individual lineage than to common ancestry. None of phylogenetic signals parameters estimated were statistically significant, and there was no phylogenetic correlation between CTmax and recorded butterfly habitat. Phylogenetic signals consistent with a Brownian model of trait evolution for thermal tolerance is not straightforward, being reported in some animal groups [31], but not in others; for example, in beetles the phylogenetic relationships among species have little or no role in shaping CTmax [8].

Butterfly species showed differences in thermal tolerance despite the habitat of occurrence, and formed two statistically significant groups: the resistant group and the non-resistant general group. The resistant group was composed mostly of species that occur in the forest islands (six from eight species occurred in the majority in this habitat). From this group, one genus (*Hermeuptychia*) and one species (*Y. renata*) are known to prefer canopy strata to the understorey [32], so they live in a stratum with high solar incidence [13].

When we evaluated the CTmax of the species between habitats, the butterflies present in the campo rupestre showed higher thermal tolerance. The campo rupestre is associated with intense solar incidence [15] so species occurring in this habitat face high temperatures during the day. The CTmax of the campo rupestre species and the forest islands did not differ when comparing the mean critical temperature for each species. In general, communities are driven by the presence of abundant species, the “biomass-ratio hypothesis” [33]. We suggest that species *Y. patricia* and *G. muscosa* (the most dominant species in the campo rupestre and the forest islands, respectively) are influencing the thermal pattern of their environments, where they are more abundant.

When considering only the campo rupestre, there was no difference between the thermal resistances of the species (53.1 ± 8.0 °C). This similarity may be associated with the spatial homogeneity of the habitat: it is an open environment with reduced stratification, and mostly there is no shade or more damp areas [12]. A different pattern was observed in the forest islands: two groups of species with significantly different means of CTmax, the resistant group (52.8 ± 7.0 °C), and the non-resistant group (47.4 ± 6.5 °C). The forest islands provide at least two strata: the canopy and the understorey [13,34]. The canopy can reduce high solar incidence at soil level, sunny and shaded areas can be observed inside the forest, and the existence of the canopy can, therefore, separate the butterfly community into two different communities [34]. This vegetative structure produces a heterogeneous thermal environment, and the areas that receive lower solar incidence are cooler [34]. A heterogeneous thermal environment also provides diverse microhabitats, and this can influence the higher thermal amplitude of the community that exhibits species with different thermal tolerances [35,36]. Consequently, species from the forest islands display heterogeneity in thermal tolerance; for instance, *P. grimon* species have a lower thermal tolerance (40 °C) and *H. epinome* have a higher one (52.8 ± 4.3 °C).

## 5. Conclusions

This is the first study to provide evidence of the influence of environmental conditions on the thermal tolerance of fruit-feeding butterflies. In summary, our results suggest that this is an evolutionary adaptation of the species, and habitat does not change the tolerance but can select the species, acting as environmental filters. More heterogeneous habitats have multiple microhabitats that can hold species with different temperature tolerances.

## Figures and Tables

**Figure 1 insects-11-00278-f001:**
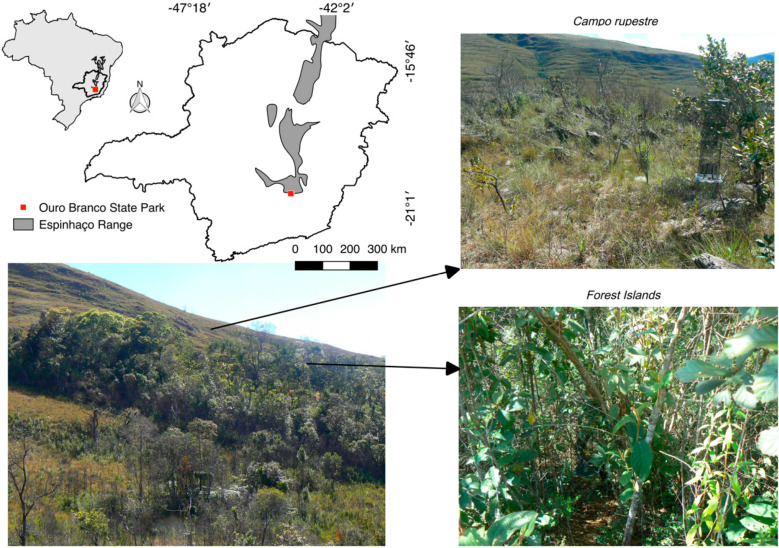
Mountaintop environment at Espinhaço mountain range from Serra do Ouro Branco State Park (20°31′ S, 43°41′ W) studied.

**Figure 2 insects-11-00278-f002:**
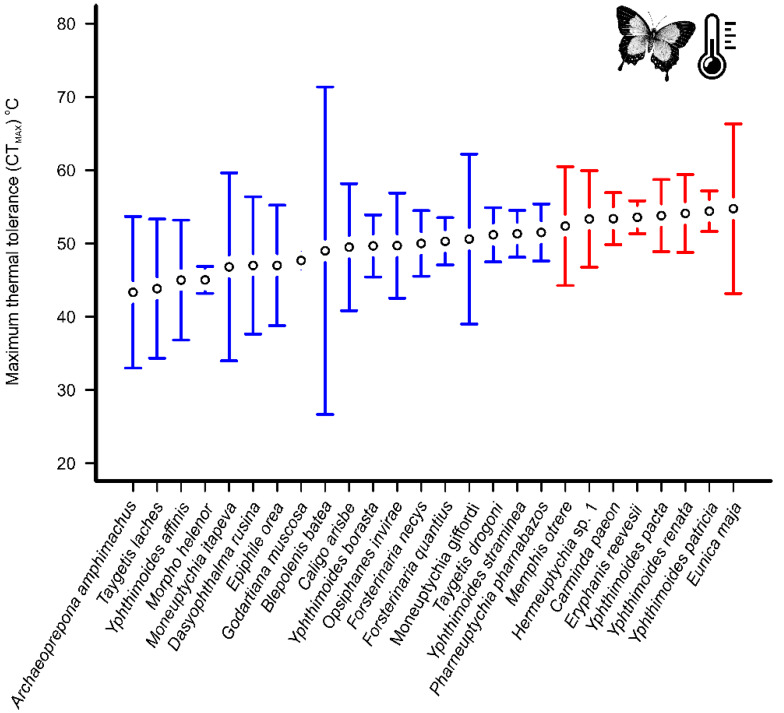
Mean of maximum critical temperature (CTmax) in °C per specimen of fruit-feeding butterflies sampled in the mountaintop environments. Species with higher thermal tolerances are shown in red. The bars represent 95% confidence interval.

**Figure 3 insects-11-00278-f003:**
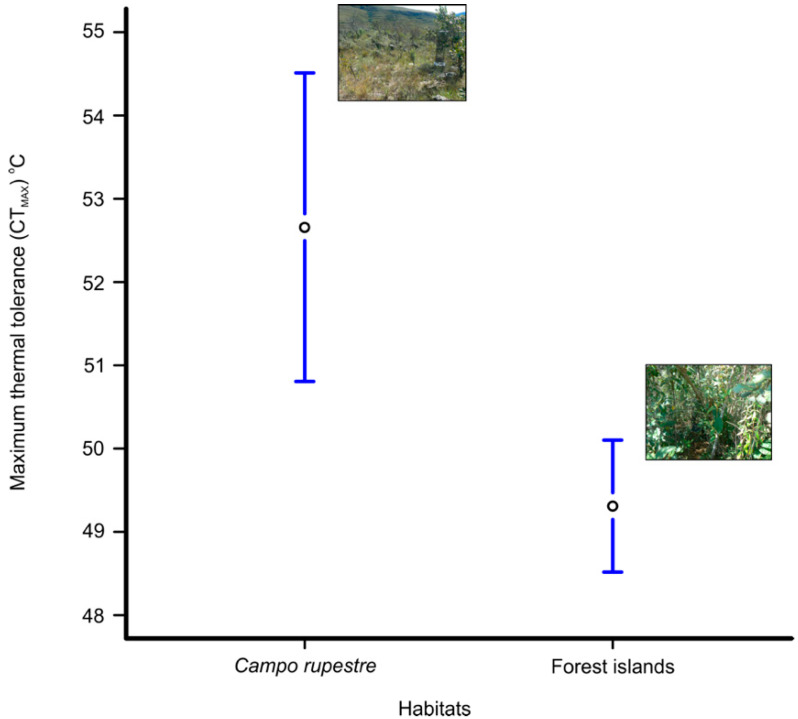
Mean of maximum critical temperature (CTmax) of the fruit-feeding butterflies sampled in each mountaintop environment: forest islands versus campo rupestre. Only species with more than three specimens were used in this analysis. The bars represent 95% confidence interval.

**Figure 4 insects-11-00278-f004:**
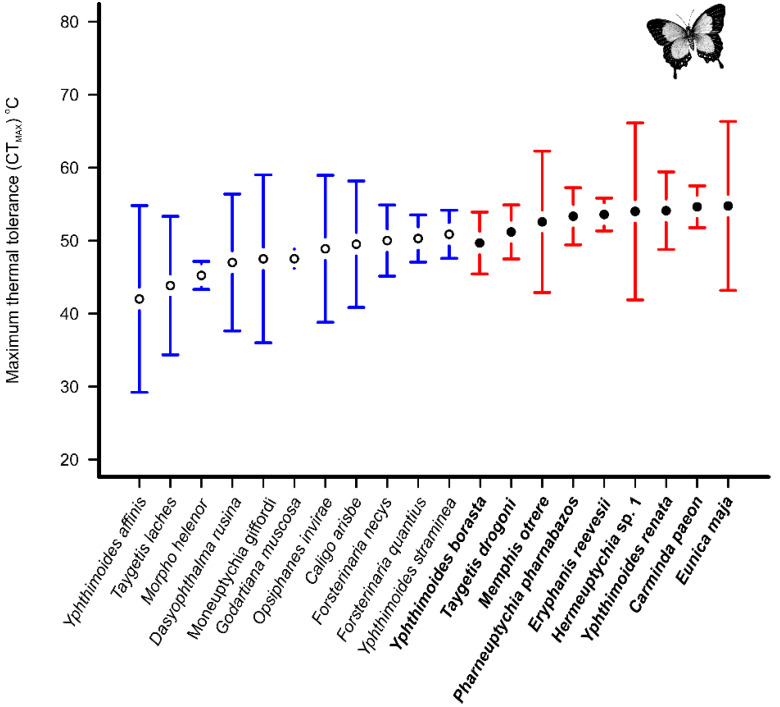
Mean of maximum critical temperature (CTmax) of fruit-feeding butterfly species from forest islands. The species were divided in two groups with high (red) and low (blue) CTmax. Only species with more than three specimens were used in this analysis. The bars represent 95% confidence interval.

**Figure 5 insects-11-00278-f005:**
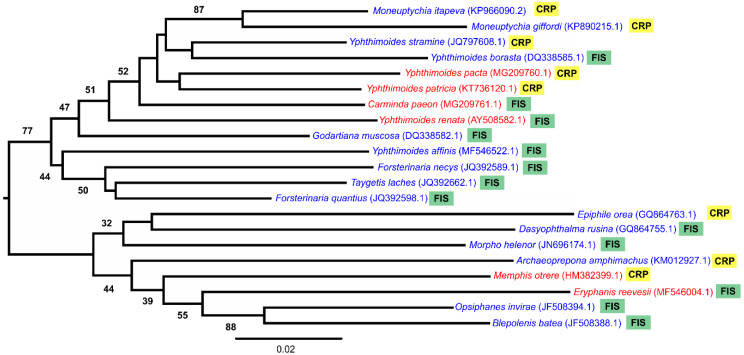
Neighbor joining (NJ) tree showing the phylogenetic relationship of fruit-feeding butterflies with distinct CTmax and habitat occurrence. In red, species with higher thermal tolerance, and in blue, species with lower thermal tolerance, followed by habitat occurrence. CRP = campo rupestre and FIS = forest islands. GenBank accession number inside brackets at tips; numbers above/below branches are bootstrap values.

**Table 1 insects-11-00278-t001:** List of butterfly species tested for thermal tolerance in Serra do Ouro Branco State Park, with the mean thermal tolerance and the habitat that occurred.

Species	Maximum Thermal Tolerance (Mean ± SD) in °C	Habitats		Total
		Campo Rupestre	Forest Islands	
*Archaeoprepona amphimachus* (Fabricius, 1775)	43.3 ± 4.2	2	1	3
*Blepolenis batea* (Hübner, 1821)	49.0 ± 9	1	2	3
*Caligo arisbe* (Hübner, 1822)	49.5 ± 5.4	0	4	4
*Carminda paeon* (Godart, 1824)	53.4 ± 7.8	2	19	21
*Colobura dirce* (Linnaeus, 1758)	52 ± 2.8	0	2	2
*Dasyophthalma rusina* (Godart, 1824)	47 ± 7.5	0	5	5
*Doxocopa laurentia* (Godart, 1824)	59.0	1	0	1
*Epiphile orea* (Hübner, 1823)	47 ± 5.2	4	0	4
*Eryphanis reevesii* (Doubleday, 1849)	53.6 ± 3.7	0	13	13
*Eunica cuvierii* (Godart, 1819)	60.5 ± 2.1	2	0	2
*Eunica maja* (Fabricius, 1775)	54.8 ± 7.3	0	4	4
*Eunica tatila* (Herrich-Schäffer, 1855)	53	1	0	1
*Forsterinaria necys* (Godart, 1824)	50 ± 7.7	1	13	14
*Forsterinaria quantius* (Godart, 1824)	50.3 ± 5.6	0	14	14
*Godartiana muscosa* (Butler, 1870)	47.7 ± 6.5	4	94	98
*Hamadryas epinome* (Felder and Felder, 1867)	53 ± 4.2	0	2	2
*Hamdryas februa* (Linnaeus, 1758)	48	0	1	1
*Hermeuptychia* sp1 (Forster, 1964)	53.5 ± 6.3	2	4	6
*Historis odius* (Fabricius, 1775)	33.0	1	0	1
*Junonia evarete* (Cramer, 1779)	69.0	1	0	1
*Memphis acidalia* (Hübner, 1819)	47.0	1	0	1
*Memphis appias* (Hübner, 1825)	64.0	1	0	1
*Memphis moruus* (Fabricius, 1775)	54.5 ± 2.1	1	1	2
*Memphis otrere* (Hübner, 1825)	52.4 ± 9.7	1	7	8
*Moneuptychia giffordi* (Freitas, Emery, and Mielke, 2010)	50.6 ± 9.3	1	4	5
*Moneuptychia itapeva* (Freitas, 2007)	46.8 ± 10.3	4	1	5
*Moneuptychia walhbergi* (Freitas, Barbosa, Siewert and Mielke, 2015)	54 ± 1.4	0	2	2
*Morpho helenor* (Cramer, 1776)	45 ± 5.6	2	36	38
*Opoptera sulcius* (Staudinger, 1887)	39 ± 9.9	0	2	2
*Opsiphanes invirae* (Hübner, 1808)	49.7 ± 5.8	1	4	5
*Paryphthimoides grimon* (Godart, 1824)	40.0	0	1	1
*Paryphthimoides poltys* (Prittwitz, 1865)	47.0	0	1	1
*Pharneuptychia pharnabazos* (Bryk, 1953)	51.5 ± 4.7	2	6	8
*Taygetis drogoni* (Siewert, Zacca, Dias and Freitas 2013)	51.2 ± 5.5	0	11	11
*Taygetis laches* (Fabricius, 1793)	43.4 ± 3.8	0	3	3
*Yphthimoides affinis* (Butler, 1867)	45 ± 7.8	2	4	6
*Yphthimoides angularis* (Butler, 1867)	58.5 ± 6.4	2	0	2
*Yphthimoides borasta* (Schaus, 1902)	49.7 ± 5.5	0	9	9
*Yphthimoides pacta* (Weymer, 1911)	53.8 ± 4.0	4	1	5
*Yphthimoides patricia* (Hayward, 1957)	54.4 ± 8.1	34	1	35
*Yphthimoides renata* (Stoll, 1780)	54.1 ± 7.4	0	10	10
*Yphthimoides* sp1	64	1	0	1
*Yphthimoides* ps2	54	1	0	1
*Yphthimoides straminea* (Butler, 1867)	51.3 ± 8.7	2	29	31
*Zaretis isidora* (Cramer, 1779)	57.0	0	1	1

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
