# Peer review of "Thermal Tolerance of Fruit-Feeding Butterflies (Lepidoptera: Nymphalidae) in Contrasting Mountaintop Environments"

_insects, 2020, doi:10.3390/insects11050278_

Round 1

Reviewer 1 Report

General comments

This is a very interesting study that tries to connect species thermal tolerance to specific habitat types and explore further any evolutionary patterns that might arise. The sampling is robust, but the statistical approach raises some concerns. Below, I provide my specific comments that could be used for the improvement of the manuscript.  

Specific comments

Abstract

Line 13 Please delete possible

Introduction

Lines 32-34: Please consider dividing the sentence.

Lines 41-42: Please rewrite for clarity

Line 46: Better use host plants. In general, in Lines 45-48 you started with butterfly ecological characteristics and ended up with the statement of different butterfly assemblages among different habitats. All these are true, but I would consider rewriting for clarity.

Line 52: You might want to delete the references into the parenthesis.

Line 54: Please replace the reference to a number

Line 57: same as before, reference needs to be replaces by an integer

Line 58: Please use capital in the beginning of the sentence

Line 64: It might be useful to give an explanation. Why do you expect this differentiation in thermal tolerance?

Materials and Methods

Lines 96-98: It would be very useful to explain further the experimental set up. In addition, it would be very helpful to give a more detailed explanation of the CTmax (e.g., temperature at which motor control is lost in animals).

Lines 103-121: Overall, I don’t agree with this multiple model’s approach, I think you could combine more than one research questions and fit them all in one model. For example, you used three models (lines 103-112) when the response variable is always the same. Why can’t you use more than one explanatory variables, like for example species and habitat type and also test for their interaction. You could avoid running two different models because you have an unbalanced design if you put species as a random effect. In addition, I am not sure I understand what the response variable at species level is. Is it the average of the individuals you measured for each species?

Lines 113-120: I think you make the same mistake of applying many models when in fact you could combine all you research questions by using only one, more complex though. Please see my previous comment.

Finally, to be able to say that some species have higher thermal tolerance than others wouldn’t you need post hoc tests?

Line 126: marker?

Results

Line 149: Please delete “This”.

Figure 2: No red is shown in the graph

Line 163: Don’t you mean individuals?

Line 164: this replies to my previous comment about species

Figure5: It would be easier to read if the habitats had a lighter color or even without a colored background.

Discussion

Line 230: I would suggest avoiding such statements.

Line 252-253: To my understanding species with a mean CTmax > 50°C consider to be species with high tolerance compared to the ones with mean CTmax < 50°C. However, there is no clear reference in the MS.

233-238: Why do you think you found no differentiation between habitat types at species level? Do you think a different approach where both individual and species level information would be included would support otherwise?

Line 240: if G.muscosa is the only species found in both environments I am not sure how you ran your model (lines 113-116).

Author Response

Reviewer #1

  • Line 13 Please delete possible

Thanks, we accepted the change.

  • Lines 32-34: Please consider dividing the sentence.

Instead of: Ectotherms, such as insects, present physiological characteristics that make them extremely temperature dependent [1] making them good models for studies that seek to predict organismal responses to possible global climate change and habitat choice, we changed to: Ectotherms, such as insects, present physiological characteristics that make them extremely temperature-dependent [1]. This limitation makes them good models for studies that seek to predict organismal responses to possible global climate change and habitat choice.

  • Lines 41-42: Please rewrite for clarity

Instead of: Additionally, estimating the fluctuations in temperatures experienced by species in several microhabitats is a difficult task [6]. We changed to: Additionally, estimating the temperature limits that species can tolerate in several microhabitats is a difficult task.

  • Line 46: Better use host plants.

Thanks for noticing

  • In general, in Lines 45- 48 you started with butterfly ecological characteristics and ended up with the statement of different butterfly assemblages among different habitats. All these are true, but I would consider rewriting for clarity.

We split the paragraph in two. And we changed: Butterflies are very sensitive to abiotic conditions, and can only occur where the host plants exist, therefore closely related to their habitat, and since they respond rapidly to changes in the environment they are a good bioindicator organism [9-11], thus different habitats are expected to contain different butterfly species. To: Butterflies are very sensitive to abiotic conditions such as temperature, humidity, and presence of shades or light (), as well as the biotic because they depend on the presence of the host plants. Consequently, closely related to their habitat, and since they respond rapidly to changes in the environment, they are useful bioindicator organisms [9-11]. Thus different habitats are expected to contain different butterfly species.

  • Line 52: You might want to delete the references into the parenthesis.

Deleted

  • Line 54: Please replace the reference to a number Line 57: same as before, reference needs to be replaces by an integer

We revised all the references, thanks.

  • Line 58: Please use capital in the beginning of the sentence

Thanks for noticing

  • Line 64: It might be useful to give an explanation. Why do you expect this differentiation in thermal tolerance?

We inserted in line (x) one possible explanation: “The consequence of this difference in the soil led to a differentiation of vegetation, leading to a differentiation of habitat, one a grassland and the other forest. Making the campo rupestre and the forest islands two completely different habitats, one more exposed to the sun, winds, and dry and the other with less variation in temperature and humidity.”. And in line 64 we added:  We expect that the butterfly species of the campo rupestre exhibit higher thermal tolerance values than the butterfly species of the forest islands, because the temperature and the sun incidence is more prominent in campo rupestre.

  • Lines 96-98: It would be very useful to explain further the experimental set up. In addition, it would be very helpful to give a more detailed explanation of the CTmax (e.g., temperature at which motor control is lost in animals).

We inserted some more information. The sentence is: “The temperature inside the glass was tracked using a thermometer. When the butterflies lost motor control, usually not staying up, falling aside, the temperature of the air inside the glass was recorded and this was considered the CTmax [see 3, 8] of the individual tested.”

  • Lines 103-121: Overall, I don’t agree with this multiple model’s approach, I think you could combine more than one research questions and fit them all in one model. For example, you used three models (lines 103-112) when the response variable is always the Why can’t you use more than one explanatory variables, like for example species and habitat type and also test for their interaction.  You could avoid running two different models because you have an unbalanced design if you put species as a random effect. In addition, I am not sure I understand what the response variable at species level is. Is it the average of the individuals you measured for each species? –

Ok, we built a linear mixed model as you suggested. We changed the analysis to: To test if the thermal tolerance of butterfly species and the habitat differ, we built a linear mixed model (LMM) with the CTmax of each tested specimen (response variable) and the species, the habitat and the interaction between them (explanatory variable). Because we tested some species more than others, we inserted the species as the random variable. For this model, we only used species with more than three individuals tested for CTmax. We built a null model to contrast the first model. If the models differ, we assumed that the thermal tolerance is different in at least one of the response variables. For the model simplification, we extracted first the interaction and then contrasted with our first model. If significant, the first model was considered the final model, and if not significant, we assumed the minimal model without the interaction. For the mixed model we used the package lme4 [22].

  • Lines 113-120: I think you make the same mistake of applying many models when in fact you could combine all you research questions by using only one, more complex though. Please see my previous comment.

In the three last analyses, the response variable was the same, but from different data, so the models had to be different. In the first, we used species that occurred in both habitats (only one species was tested more than three times in each habitat), in the second we used only species tested at least three times only in campo rupestre, and in the last species tested at least three times in the forest island.

  • Finally, to be able to say that some species have higher thermal tolerance than others wouldn’t you need post hoc tests?

We forgot to mention, thanks for noticing. We inserted the phrase: If the models were significant, we did a contrast analysis to find how many groups the species were separated and which species belonged to each group.

  • Line 126: marker?

Yes, done!

  • Line 149: Please delete “This”.

Thanks for noticing, changed!

  • Figure 2: No red is shown in the graph

Thanks for noticing, changed!

  • Line 163: Don’t you mean individuals?
  • Line 164: this replies to my previous comment about species

Ok

  • Figure5: It would be easier to read if the habitats had a lighter color or even without a colored background.

Thanks, the suggestion was accepted

  • Line 230: I would suggest avoiding such statements.

We excluded the sentence

  • Line 252-253: To my understanding species with a mean CTmax >50°C consider to be species with high tolerance compared to the  ones with mean CTmax < 50°C. However, there is no clear reference in the MS.

No, the contrast between species showed that the species were separated into two groups according to the CTmax. We inserted in the material and methods that we did a contrast test to see the groups once the test was significant.

  • 233-238: Why do you think you found no differentiation between habitat types at species level? Do you think a different approach where both individual and species level information would be included would support otherwise?

We found differentiation only in the forest habitat. We think that the campo rupestre is more homogeneous that the forest. The explanation for the differentiation in the forest habitat is that has many microhabitats so can hold species with sunny and shadow preferences, for exemple.

  • Line 240: if muscosa is the only species found in both environments I am not sure how you ran your model B (lines 113-116).

We only used the data of the CTmax of the species G. muscosa. We expected more species tested at least three times in each habitat, but it didn’t happened. In the Materials and methods we explained: “Then, to test for different thermal tolerance between species that occur in both habitats, we built a GLM with the CTmax of the species that occurred in both habitats (response variable) and the habitats (explanatory variable). For this analysis, we considered only species where more than three individuals had been tested for thermal tolerance in each habitat. Considering the reviewer questions we decided to remove this model due the difference between the number of butterflies across habitats. 

Reviewer 2 Report

Overall the study was quite sound and the results are indeed interesting, pointing to evolutionary adaptation of the various species. It would have been equally interesting to measure other key traits such as melanism, thorasic volume/body size, sex, etc. to may play a role -but realize this was outside the scope of the research.  I do suggest that more detail is needed in the methods, specifically for the CTMax testing. Please see comments on the manuscript.

Author Response

Reviewer #2

  • It would be good to explain more how the individual traps were deployed (e.g. at what height, orientation within the site, etc). This would be useful in order to replicante the study.

We sampled fruit-feeding butterflies from three sites in the campo rupestre (open) and three sites in the forest islands (shaded). The sites were at least 500m distant from each other. At each site five butterfly traps were installed at a distance of 50 m from each other in a line, and with the base about 1m above the ground, in total there were 30 traps. In the forest islands, the traps were hanged more in the middle of the fragment, to avoid the edge, and the trap line followed the shape of the fragment. The butterflies were collected with attractive bait traps using fermented banana. The traps were inspected every day during the sampling period. All captured butterflies were marked with an identification number and kept in a glass container (52 x 27 x 39 cm) without the top, closed with a thin fabric and with some cotton emerged with water in the bottom to reduce the stress during the transportation to the laboratory to carry out the thermal tolerance tests. The number in the butterflies was related to the date of capture, the number of the trap, the fragment, and the habitat. Fieldwork was conducted monthly over a year (December 2017 to November 2018) in five days of sampling.

No, the glass box was only covered with some light fabric and some cotton balls with water were inside the box. 

  • What metrics were collection on each specimen? Was there an indicator of age such as based on wing wear?

We did not use the age, the metrics were only the place captured, the date and the species.

  • What brand, model, accuracy?

We inserted: Brand quimis, model G215M1, precision +- 0,05°C.

  • This process and indicator (lost of flight) needs to be more fully explained. Did all butterflies attempt to fly in the 2L glass?
  • We inserted more information. The sentence is: “The temperature inside the glass was tracked using a thermometer. When the butterflies lost motor control, usually not staying up, falling aside, the temperature of the air inside the glass was recorded and this was considered the CTmax [see 3, 8] of the individual tested.”

Round 2

Reviewer 1 Report

All requested changes were followed nicely by the authors.

I have no further comments.